# A Controlled Experiment to Test the Efficacy of Ground-Penetrating Radar in the Search for Clandestine Burials in Poland

Tomasz Borkowski [1,*], Fernando Constantino [2], Alexandre Novo [2], Jamie Frattarelli [3] and Maciej Trzciński [4]

1    Institute of National Remembrance, 02-676 Warsaw, Poland
2    Screening Eagle Technologies, 8603 Schwerzenbach, Switzerland
3    Alta Archaeological Consultancy, Santa Rosa, CA 95403, USA
4    Department of Forensic Sciences, University of Wrocław, 50-137 Wrocław, Poland
*    Correspondence: tomasz.borkowski@ipn.gov.pl

**Abstract:** The international workshop 'Forensic Search and Recovery of Clandestine Graves' took place over two days in 2021 in Wroclaw, Poland. The goal of the workshop was to improve search methods and techniques related to the examination of clandestine burial sites. Geophysical methods were used by an international team of multi-disciplinary specialists to detect simulated burial sites. The training focused on testing methods, including Ground-Penetrating Radar (GPR), to verify the effectiveness of the methods in the search for two features representing clandestine burials. The forensic community in Central European countries, including Poland, has been slow to adopt these technologies due to controversial results. While geophysical research is successfully carried out in archaeological research and forensic contexts internationally, its application in the activities of the prosecutor's office and the police in Poland has been relatively unsuccessful. This has resulted in several controversies related primarily to the erroneous expectations of how the methods are successfully applied. This may be the result of operator inexperience in applying these methods to the search for clandestine burials. This training paired an experienced GPR operator with law enforcement teams and archaeologists, leading to the successful discovery of simulated burials using GPR.

**Keywords:** forensic archaeology; field training; GPR



## 1. Introduction

The specialist workshop Forensic Search and Recovery of Clandestine Graves was held from 20–21 September 2021 in Wrocław (Poland). Originally, the workshop was scheduled to take place in 2020, but was postponed due to the COVID-19 pandemic. The organizers were the University of Wrocław (Department of Forensic Sciences) and the Institute of National Remembrance, in cooperation with the Provincial Police Department in Wrocław. The workshop was held under the auspices of the European Network of Forensic Science Institutes (ENFSI).

The goal of the field training was to test the effectiveness of geophysical methods in the search for concealed corpses. In many countries, there is already a specialization in these methods, and the concept of forensic geophysics has become widespread in science [1,2]. GPR can be employed in criminal investigations, as it has become more accessible, efficient, and relatively lower in cost, while detecting possible buried evidence areas and eliminating other non-related areas with minimal disturbance to the ground [2].

In Poland, geophysical methods have been employed in traditional archaeological research. Successful cases include the use of GPR to map medieval ring-forts in Poland as part of the cultural landscape [3]. GPR was also applied in an urban planning context, when the method was employed to locate burials in a historic unmarked cemetery. The survey

was able to detect the location and spatial configuration of the internments [4]. Geophysics has been also applied by law enforcement in searching for victims of contemporary criminal offenses. A separate type of activity is the search for victims of the communist regime, which is carried out by the Office for Search and Identification of the Institute of National Remembrance. Under Polish law, communist crimes, as well as war crimes and crimes of genocide, do not have a statute of limitations. It should be noted here that in the case of a common murder, the Polish Penal Code establishes the statute of limitations for this crime to be 30 years.

The use of geophysical methods in forensic applications has developed a poor reputation in Poland with both law enforcement agencies and the Institute of National Remembrance. The methods have generally shown a low success rate during searches for single and mass graves. The coauthors of this paper (T.B. and M.T.) participated in approximately 20 searches in which methods such as GPR and magnetometry were employed. The anomalies indicated by these devices were examined by digging test trenches or by geological drilling and did not confirm the existence of a grave in any case. These case studies may be reflective of several shortcomings in the application of the geophysical methods. The Polish police and prosecutor's office, when conducting such searches, mainly use external experts or specialists, as they do not have the appropriate equipment and trained staff. The Institute of National Remembrance also commissions these investigations with outside contractors, when searching for victims of the communist regime's crimes. The modest effectiveness of geophysical methods as practiced in actual criminal cases has resulted in the fact that these methods tend to be viewed as ineffectual by Polish authorities.

These results compare starkly with the success with which these methods are used and have been adopted in other countries, especially Western European ones, where forensic geophysics already has an established position. This may be attributed to the fact that the Polish experts deal mostly with traditional archaeological research and do not have adequate practice in searching for buried human remains. Therefore, these are limitations not so much related to the quality of the equipment, but rather to experience, which would allow, for example, the selection of the correct frequency of the GPR antenna to be adjusted to the specificity of a given soil. Another important issue is the ability to reliably interpret the anomalies revealed during the research. While forensic archaeology has been established as a legitimate field in Poland, forensic geophysics as a separate specialty has been met with skepticism [5].

The organization of a specialist workshop with the participation of experts from different countries created an excellent opportunity to share experiences. The training was organized at the police compound in Wrocław. The area in question was secured and monitored. In preparation for the workshop, geological surveys were carried out in 2017 and 2018. The survey results found that the selected area is challenging for geophysical research due to numerous layers of underground debris, including small metal objects and bricks. The debris was created by military operations at this location in 1945. During World War II, there were German barracks, which have since been removed. After the end of the war, the Soviet army resided in the area until 1990. The debris deposition creates a challenging environment for geophysics. Fallen masonry and disturbed stratigraphy may create chaotic feedback that is difficult to interpret [6–8].

GPR is a well-accepted technique for the detection of clandestine graves, since it was first successfully utilized in 1986 by Vaughn [7]. The advantages of using non-intrusive methods, such as GPR, are the preservation of the crime scene, minimal destruction of forensic evidence, and, therefore, enhancement of the possibility of reconstructing events at the scene [1]. The use of GPR in the search for burials has been previously tested in experimental settings. GPR has been used, along with other geophysical methods, to monitor simulated burials over a long period. Pringle et al. used pig cadavers buried under a variety of conditions to monitor changes in conditions and GPR reflections over six years [9]. A more recent test used burials containing human remains to simulate both mass burials and individual burials, looking at changes in data results over time [10].

Even though GPR can help as part of a forensic investigation sequence [11], it has not been yet fully adopted in forensic science in Poland, because specialized training is necessary for acquiring, processing, and interpreting the data. Although some authors have achieved successful results under controlled conditions, such as in a cemetery test area [12] and over a simulated urban clandestine grave [13], 3D GPR has been rarely applied in real forensic cases because it used to require special training for both data acquisition and 3D data processing [14,15]. In addition, rough surfaces and time are often strong limitations in homicide investigations, and, therefore, most surveys are still carried out collecting 2D line scan data. However, recent technological advancements, together with user experience (UX) improvements and subscription-based access, are making GPR technology more popular.

The following case study presents how applying an innovative workflow for collecting GPR data efficiently and then communicating the data strategically and clearly could become a standard procedure for forensic investigations.

## 2. Materials and Methods

The original workshop date was scheduled for April 2020, and preparations began in October 2019. Two features were constructed on the premises of the police compound. Feature 1 was a replica of a single pit grave, so a plastic skeleton, metal, wood, and plastic were deposited in an excavated pit (Figure 1). The pit was rectangular in plan, and its dimensions were 1.63 m in length, 0.62 m in width, and 0.80 m in depth (Figure 2). Feature 2 was a small oval pit, 0.82 m long, 0.52 m wide, and 0.55 m deep. It contained a plastic replica of a human skull, animal bones, and metal artefacts. Both pits were backfilled with material extracted during digging, made up of sand mixed with humus and a significant amount of brick rubble (Figure 3).

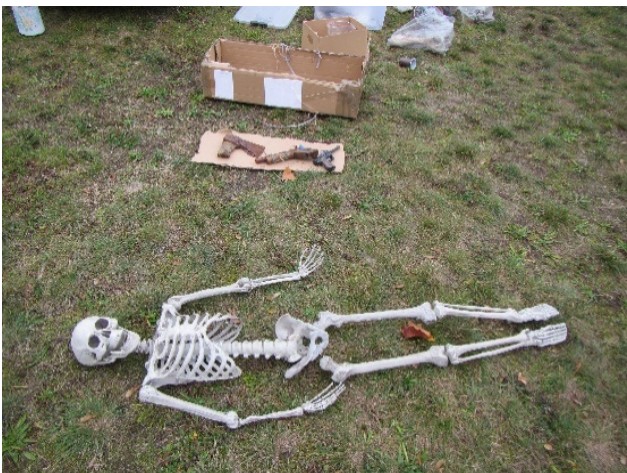

**Figure 1.** Plastic replica of human skeleton and artefacts ready to be buried. Wrocław, Poland, police compound. 29 October 2019 (photo by T.B.).

The postponement of the workshop's date lent an additional layer of real-world conditions to the experiment. The vegetation (grass) had grown enough to almost completely mask the prepared features. After almost two years, even the 'perpetrators' themselves had certain problems identifying the location of the features. In addition, the participants of the training were provided with very little information about the features they were searching for, adding to the authenticity of the exercise.

On the first day, non-invasive research was performed. Two teams with GPRs carried out surveys (Figure 4). A magnetometer and a multifunctional camera (including thermal imaging) mounted on a drone were also used. Below, we discuss the results of the GPR research conducted by F.C.

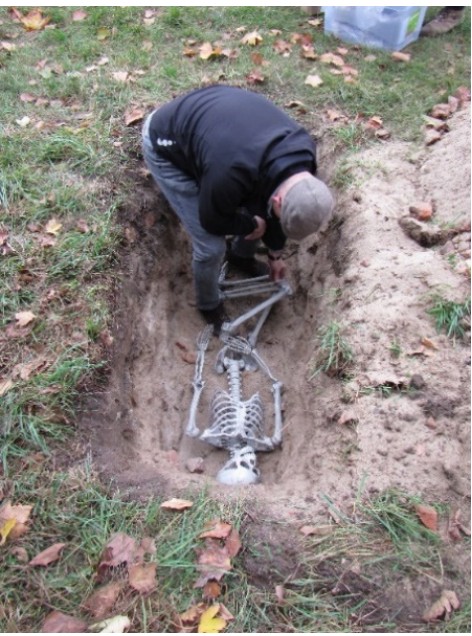

**Figure 2.** Creation of Feature 1 using plastic replica of human skeleton and artefacts. Wrocław, Poland, police compound. 29 October 2019 (photo by T.B.).

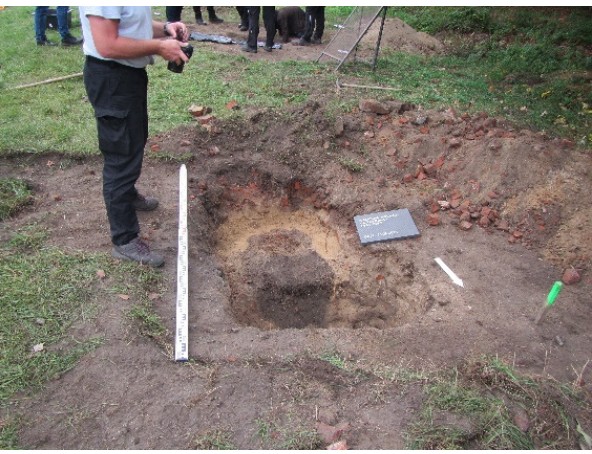

**Figure 3.** Feature 2 during excavations. Wrocław, Poland, police compound. 21 September 2021 (photo by T.B.).

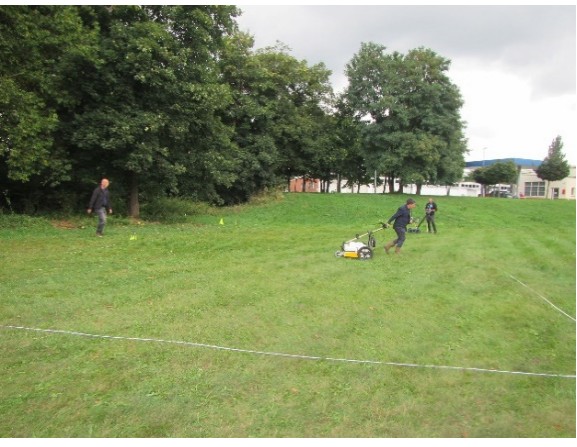

**Figure 4.** GPR survey. Wrocław, Poland, police compound. 20 September 2021 (photo by T.B.).

On the second day, excavations were carried out on two of the eight anomalies identified by the GPR, to assess their actual characteristics. Unfortunately, there was not enough time to examine all of the anomalies, so only the two most likely candidates were excavated. Further excavation could provide interesting comparative data regarding the causes of the additional anomalies. The teams decided to focus on two anomalies, which were designated as 'possible burial location 5' (Feature 1) and 'possible burial location 7' (Feature 2). An international team of archaeologists and police efficiently conducted the excavations, correctly interpreting the shapes of the features and expertly exploring their fills (Figures 5 and 6).

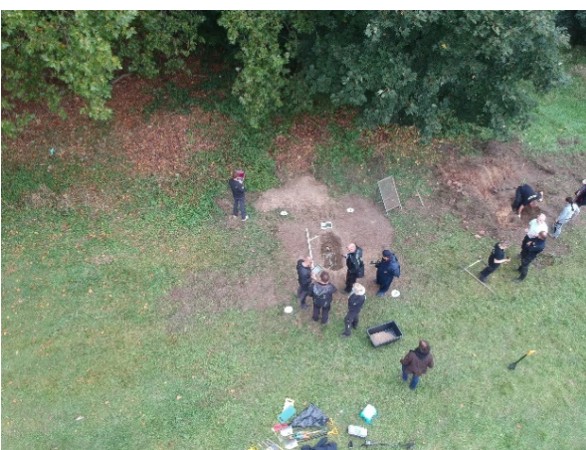

**Figure 5.** Feature 1 (possible burial location 5) and Feature 2 (possible burial location 7) during excavations. Wrocław, Poland, police compound. 21 September 2021 (photo by T.B.).

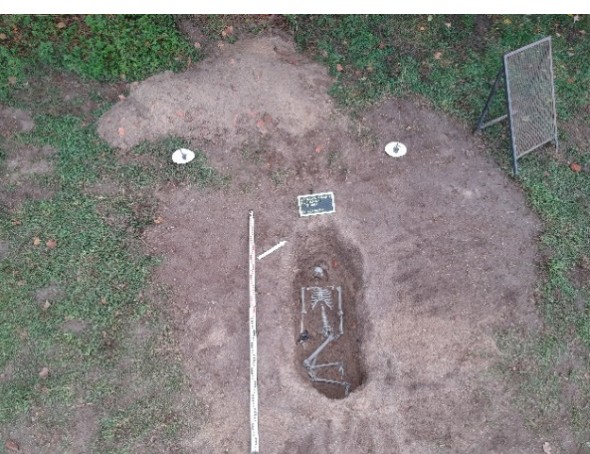

**Figure 6.** Feature 1 (possible burial location 5) during excavations with unearthed plastic skeleton. Wrocław, Poland, police compound. 21 September 2021 (photo by T.B.).

As part of a hands-on forensics field exercise, several geophysical techniques, including GPR, were used to locate clandestine graves.

The test site (Figure 7), terrain conditions, and location of burials were completely unknown to the GPR team. Most decisions regarding the clandestine grave search needed to be made on site, to keep it as close as possible to a real case scenario.

The entire area of 20 m by 15 m was scanned using a GPR system, Proceq GS8000. The wireless, subsurface mapping system was composed of: a stepped-frequency continuous wave (SFCW) GPR with a 40–3440 MHz modulated frequency range, a GNSS, and a tablet. The GS 2.0 app was used for data acquisition and real-time B-scan (radargram) and C-scan (time-slice) visualization. A grid had been set up for the other geophysical surveys, and the GPR data were collected following that same grid, for comparison consistency.

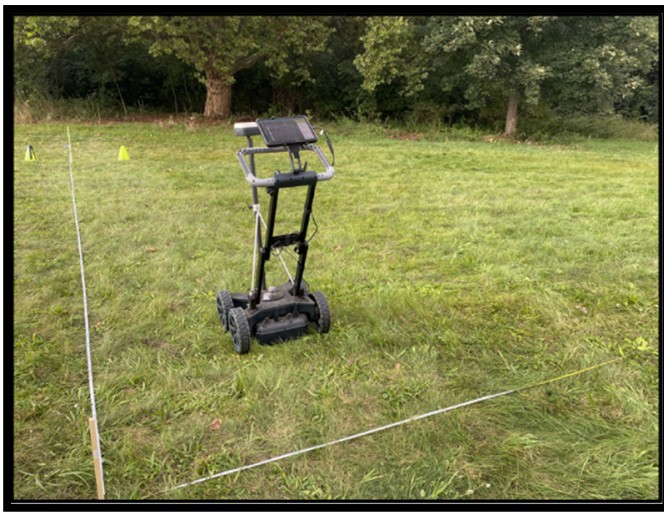

**Figure 7.** Terrain conditions scanned with the GS8000. Wrocław, Poland, police compound 20 September 2021 (photo by F.C.).

A total of 81 transect line scans were collected to cover the whole area following a path over Y direction, with a 25 cm crossline spacing and 1 cm inline spacing, using a time window of 40 ns and 650 samples per scan. An MA8000 GNSS receiver ensured centimetre-accurate positioning of all lines using built-in SSR augmentation corrections capabilities.

Some possible locations of burials were visible in real-time data (Figure 8), but it was agreed that the final results should be presented only after careful analysis and accurate marking of the graves' locations, so that the excavation teams could start their part of the exercise the day after the GPR survey.

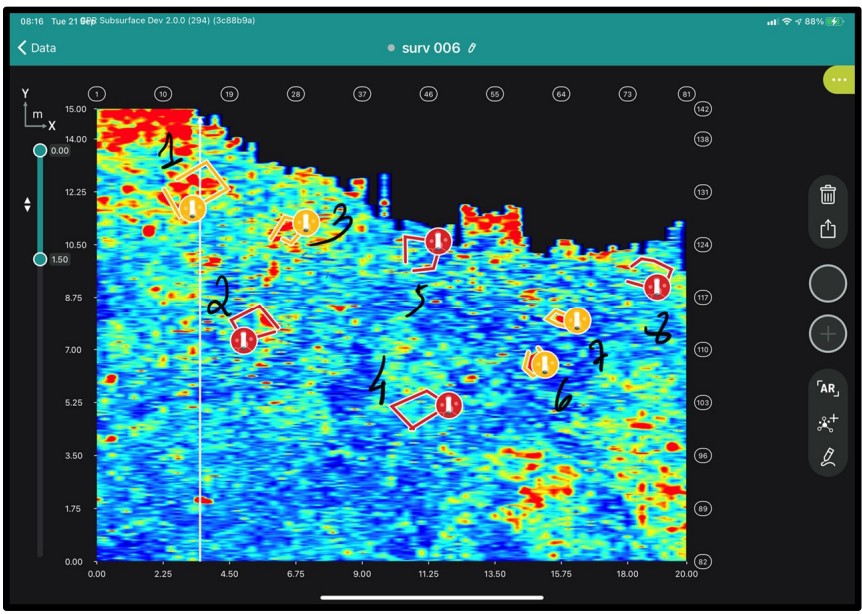

**Figure 8.** Real time C-scan showing 3D slice map, 1.5 m below the surface. The anomalies were interpreted and manual notations were made marking potential burial locations on-site. Wrocław, Poland, police compound. 20 September 2021 (edited by F.C.).

Therefore, after surveying, the data were automatically uploaded to a cloud-based web application for further data management and analytics.

### 2.1. Data Interpretation and Reporting

In the survey area, there are the remains of old building foundations. The debris from the demolished buildings was spread throughout the area, which was confirmed during excavation, and some of the wall remains could still be seen on the surface. This made it extremely difficult to distinguish GPR reflections coming from existing construction remains or from the burials. After careful data interpretation, eight possible burial locations were identified.

The GS 2.0 app was employed in the field during real-time data collection. The application allows the GPR specialist to add manual notations to the C-scans to mark anomalies in the data (Figure 8).

Indicating burial location—choosing the most likely sites to contain the burial location using augmented reality in real-time.

The possible burial locations, as determined by the GPR data, were presented to the teams, and these locations needed to be marked on the site to allow the forensic experts to do a detailed inspection of each area. The data combined with surface conditions would allow the teams to choose the most likely locations to be related to burials. Eight possible burial locations were presented to the teams, and two of them were chosen for test excavations.

Traditionally, the possible burial locations would be marked using a measuring tape relating to the area scan corners to the ground. This is standard procedure for GPR surveys over smaller areas. GPS coordinates can be used to facilitate marking possible burial sites, but this is often time-consuming. We were able to employ an augmented reality (AR) application. This feature made it possible to show the exact location of each identified burial on the surface in real-time (Figures 9–11).

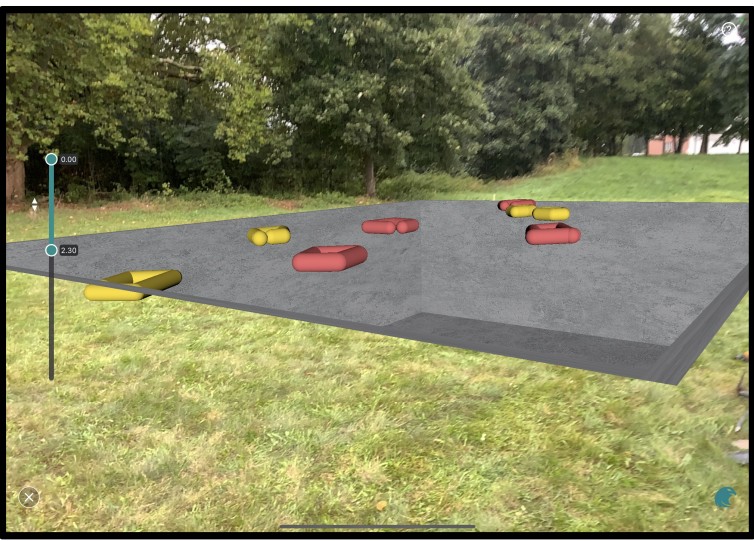

**Figure 9.** AR program projecting the positions of possible burial locations on the ground surface during the training exercise. Wrocław, Poland, police compound. 20 September 2021 (edited by F.C.).

The GPR expert used the data to show the exact physical location of the possible burial locations directly on the terrain. This allowed the data results to direct the teams to a specific location, so that the experts could evaluate the surface conditions with consideration of below-ground conditions. The use of the AR technology was reported as being very helpful in identifying the most probable areas that contained a burial (Figure 10). The ability to quickly and clearly communicate the GPR results in the field was important in allowing the teams to make an informed decision about where excavations should take place.

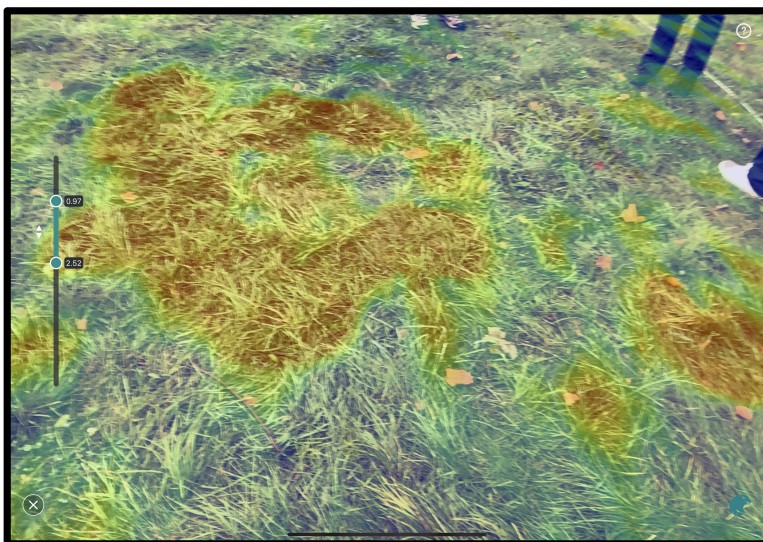

**Figure 10.** AR showing C-scan anomaly results over the ground surface in real-time. The forensic experts used the GPR projected results of possible burial locations to evaluate surface conditions and plan excavations. Wrocław, Poland, police compound. 20 September 2021 (edited by F.C.).

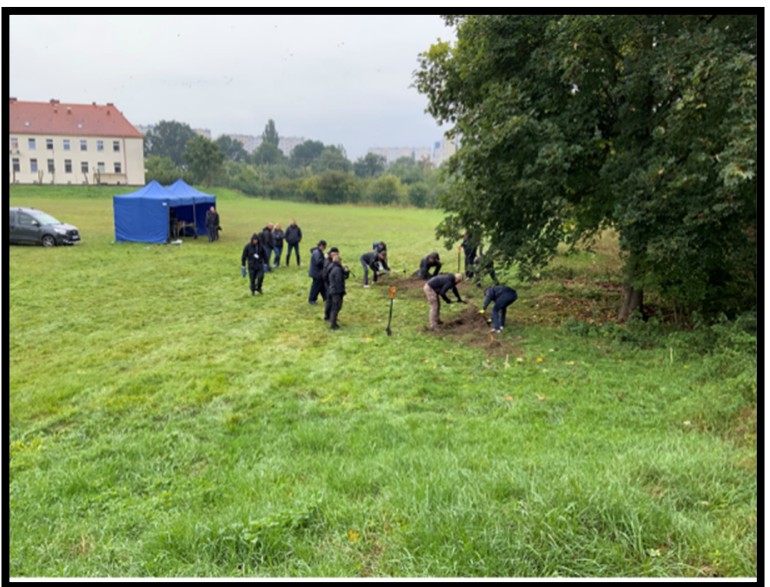

**Figure 11.** Teams excavating Feature 1 and Feature 2. Wrocław, Poland, police compound. 21 September 2021 (photo by F.C.).

*2.2. Selection of the Burial Locations to Be Excavated*

The eight locations of potential burials were marked on the ground, and each location was evaluated by experts based on visual inspections and GPR data, considering both C-scans or 3D slice maps, and B-scans or 2D transect lines. Based on this information, the teams decided to excavate possible burial locations numbers 5 and 7. Possible burial location 5 was designated Feature 1 and possible burial location 7 was designated Feature 2.

(1) Feature 1

The GPR expert determined that Feature 1 was the most likely burial site indicated in the GPR data. It is possible to see the cut in the top layer, which is a disruption in reflection patterns from the surface to 0.2 m below surface. The data then show several reflections up to 100 cm below surface (Figures 12 and 13).

(2)    Feature 2

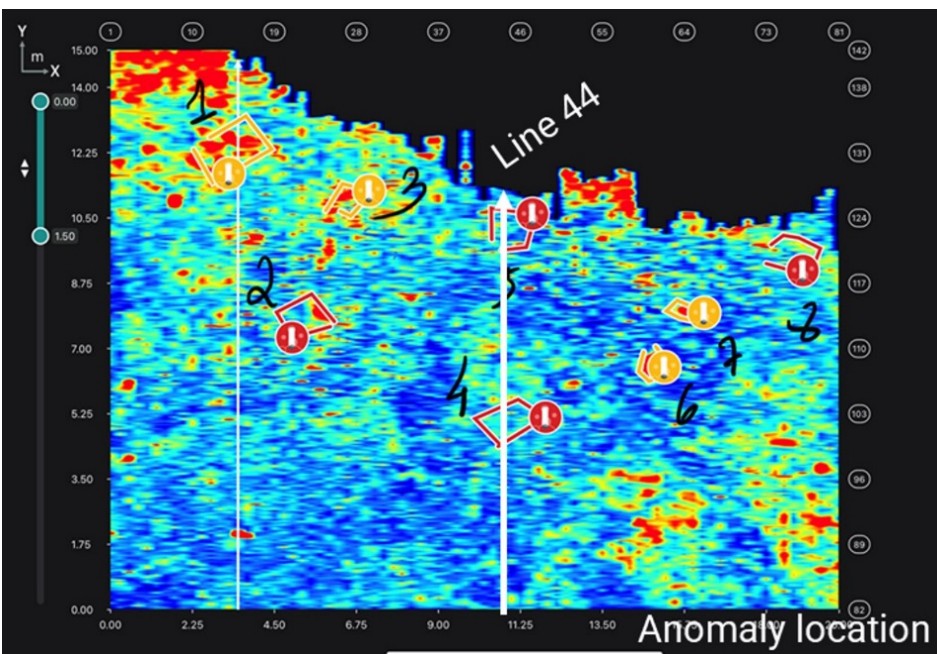

**Figure 12.** C-scan showing the 3D slice map at 1.5 m below surface depicting GPR survey transect line 44 crossing Feature 1. Wrocław, Poland, police compound. 21 September 2021 (edited by F.C.).

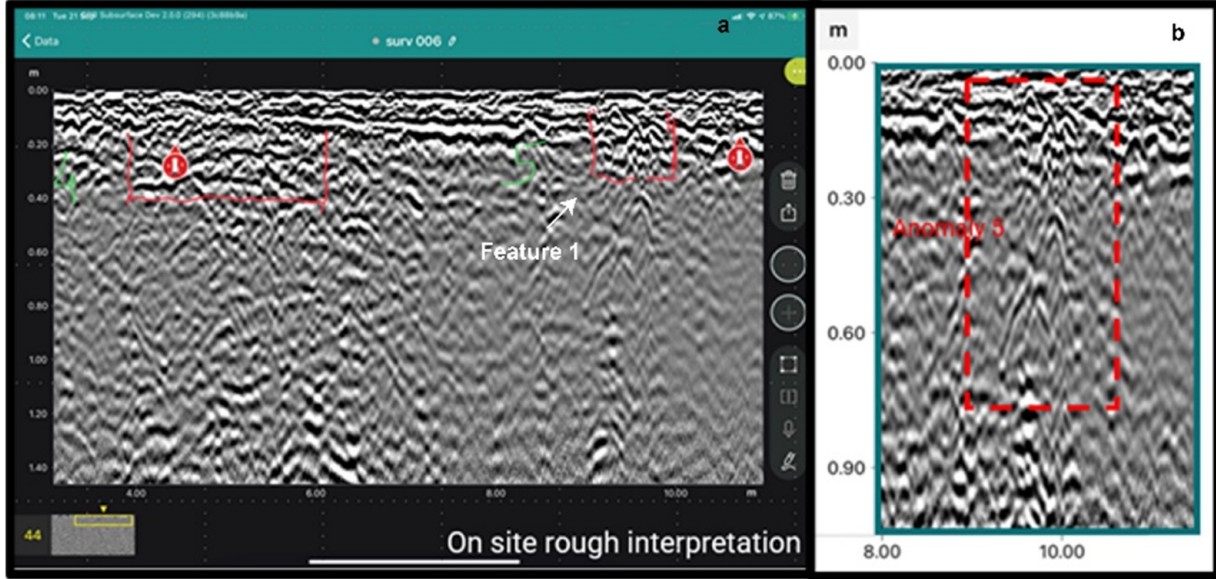

**Figure 13.** (**a**). Manual notations shown as marked in real-time data results, showing GPR data anomalies. (**b**). Detail of Feature 1 anomaly. Wrocław, Poland, police compound. 21 September 2021 (edited by F.C.).

Feature 2 shows a similar reflection pattern indicating anomalies similar to Feature 1. There are several reflections in the data, which may be attributed to debris as well. There is one area with higher amplitude that can be seen on the surface level C-scan, and it continues to be visible until the C-scan is at 1.35 m below surface (Figures 14 and 15).

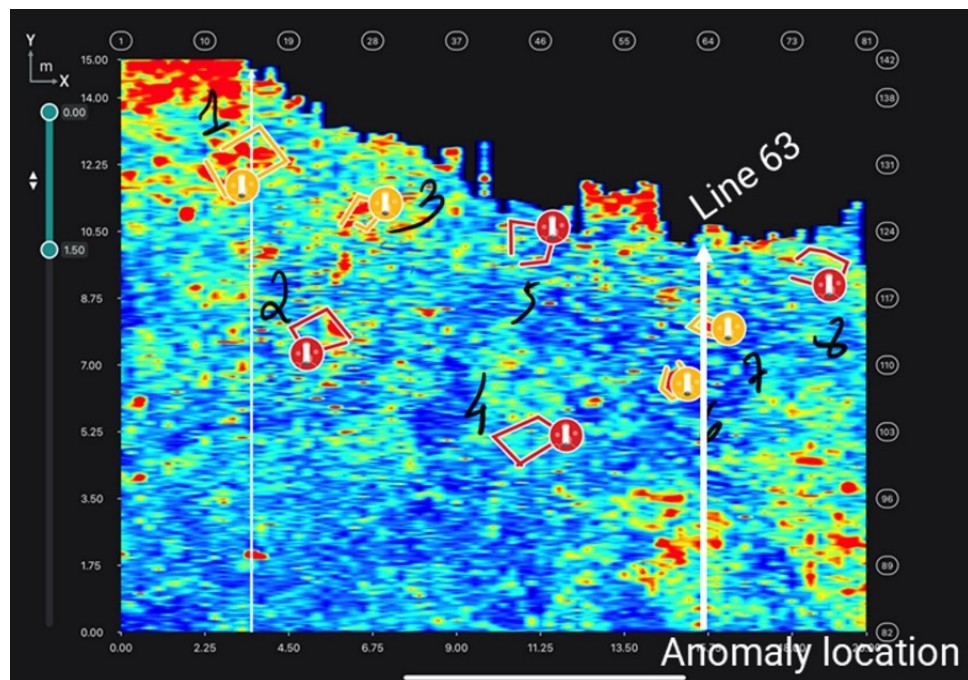

**Figure 14.** C-scan showing the 3D slice map at 1.5 m below surface depicting GPR survey transect line 63 crossing Feature 2. Wrocław, Poland, police compound. 21 September 2021 (edited by F.C.).

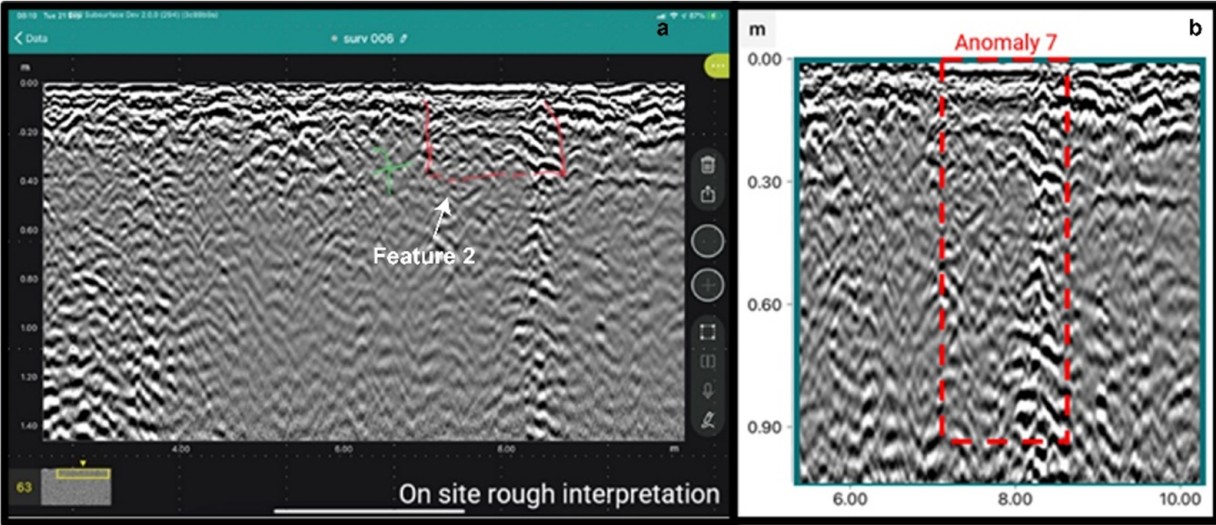

**Figure 15.** (**a**). Manual notations indicating Feature 2 in real-time data results, as identified by the GPR expert. (**b**). Post processed results of B-scan transect of Feature 2 (Anomaly 7), Wrocław, Poland, police compound. 21 September 2021 (edited by F.C.).

## 2.3. Excavation—Feature 1 (Possible Burial Location 5)

Possible burial location 5 was renamed Feature 1. The burial site was easily found by the experts' team after removing the top layer of vegetation. The edges of the burial were distinct, due to a change in soil colouration at approximately 20 cm below surface (Figure 16). The perpetrators filled the grave with the same material that was removed, and many bricks were included in the fill. A portion of the skeleton was found first, located during excavation at about 0.6 m depth (Figure 17). After the first burial material was found, two GPR transects were recorded to sample cross-sections of the burial reflections.

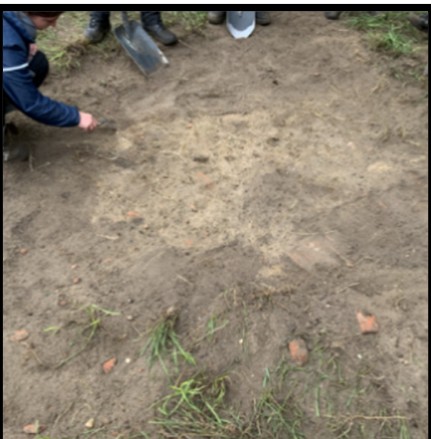

**Figure 16.** Edges of the Feature 1 burial exposed by the training team. Wrocław, Poland, police compound. 21 September 2021 (photo by F.C.).

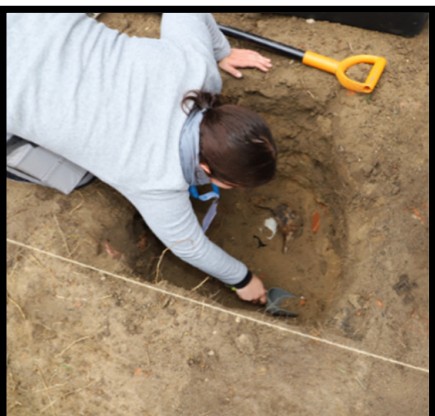

**Figure 17.** Excavation of Feature 1. Wrocław, Poland, police compound. 21 September 2021 (photo by F.C.).

Before completing the excavation of Feature 1, survey transect lines were completed with the GS8000 to identify the edges of the pit (Figure 18). The yellow lines indicate where the edges and bottom of the grave were interpreted in the data. The orange line indicated the reflections from the top of the simulated body feature (Figure 19). On GS line 1, it looks like the top of the body layer is dipping, which may be due to the position of the legs being higher than the middle of the body on the floor of the pit dug for the burial (Figure 20).

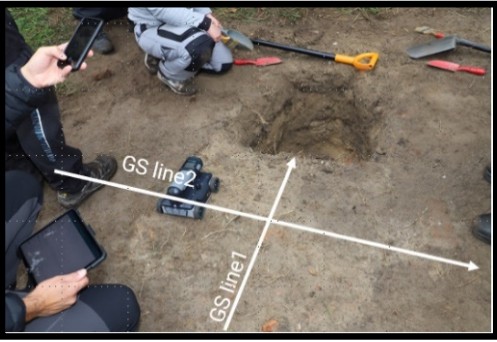

**Figure 18.** GS8000 transect lines surveyed after the removal of surface vegetation and before the completion of excavation of Feature 1. Wrocław, Poland, police compound. 21 September 2021 (photo by F.C.).

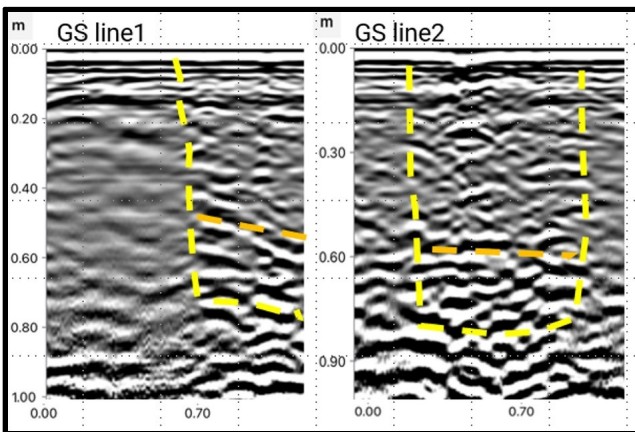

**Figure 19.** Annotated line scans of Feature 1, indicating the sides and floor of the burial pit and the top of the internment. The sloping line on top of the feature mimics the position of the skeleton. Wrocław, Poland, police compound. 21 September 2021 (edited by F.C.).

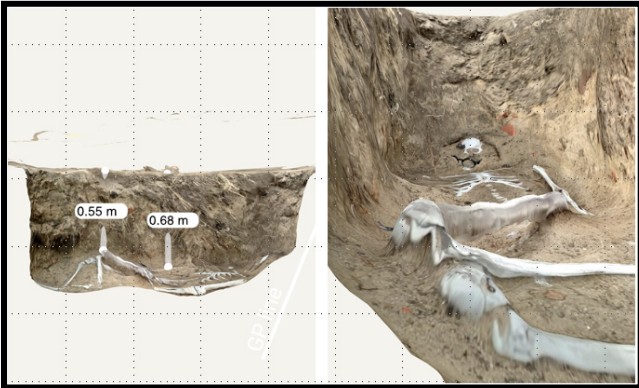

**Figure 20.** Lidar scan showing the position of the plastic skeleton in the excavation of Feature 1. Wrocław, Poland, police compound. 21 September 2021 (edited by F.C.).

*2.4. Excavation—Feature 2 (Possible Burial Location 7)*

Possible burial location 7 was renamed Feature 2. The edges of Feature 2 were less easy to identify than Feature 1. The soil was difficult to excavate and lacked the differential coloration present in Feature 1 (Figures 21 and 22). The high number of bricks in the backfill made interpretation of the GPR data difficult (Figure 23).

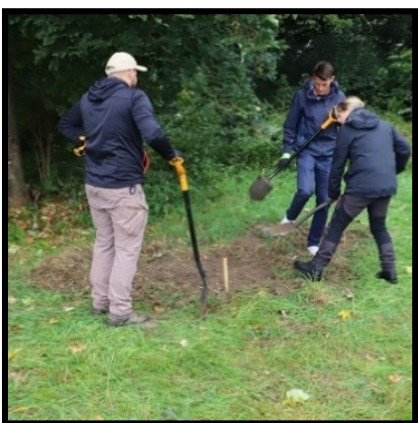

**Figure 21.** Training team looking for the edges of Feature 2. Wrocław, Poland, police compound. 21 September 2021 (photo by F.C.).

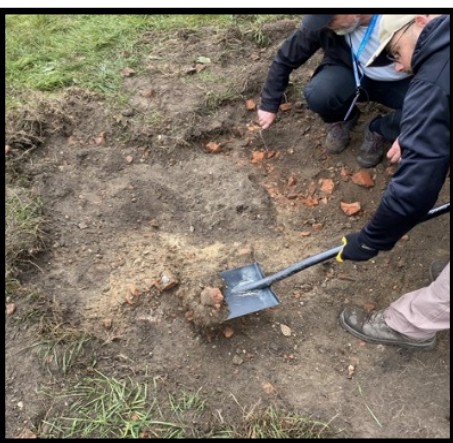

**Figure 22.** Training team looking for the edges of Feature 2. Wrocław, Poland, police compound. 21 September 2021 (photo by F.C.).

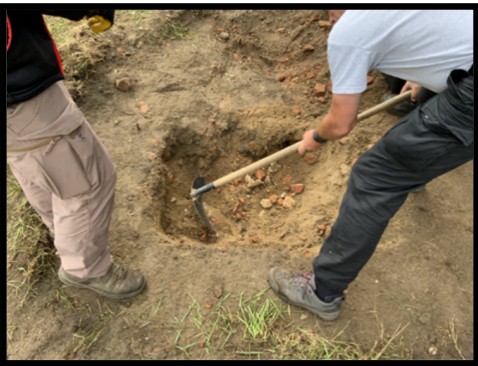

**Figure 23.** Team exposing the backfill with dense brick concentration. Wrocław, Poland, police compound. 21 September 2021 (photo by F.C.).

This grave did not contain a complete skeleton, but pieces of bones (a plastic skull and real faunal bones, including pig and horse) and metal. This type of material can create mixed reflections that are easily be confused with the reflections created by bricks (Figures 24–26).

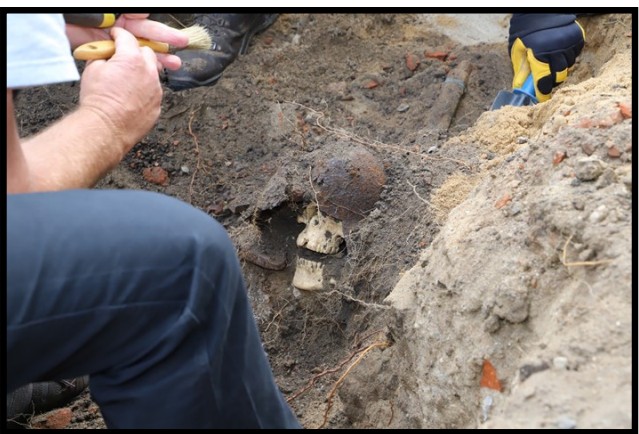

**Figure 24.** Feature 2 burial contents, plastic skeleton with metal helmet. Wrocław, Poland, police compound. 21 September 2021 (photo by F.C.).

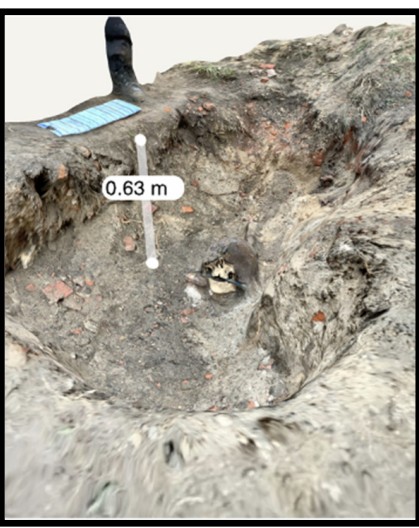

**Figure 25.** Lidar scan of Feature 2, showing the depth of the buried remains. Wrocław, Poland, police compound. 21 September 2021 (photo by F.C.).

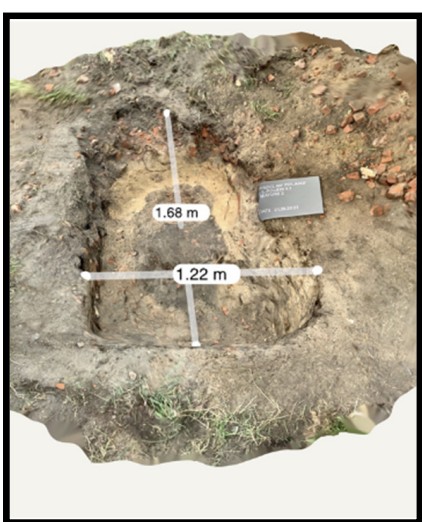

**Figure 26.** Lidar scan of Feature 2, with horizonal and vertical dimension of the remains. Wrocław, Poland, police compound. 21 September 2021 (photo by F.C.).

*2.5. Advanced Data Processing*

The GS app provides a combined view of transect line scans and slice maps in real-time, together with drawing, tagging, and export tools, to create interpretation maps in the field. Post-processing software was used for further data analytics to enhance the clarity of the survey results.

For advanced GPR data post-processing, a combination of novel web-based data analysis platforms and traditional desktop software was used. First, GPR Insights allowed for quick 3D visualization from the field, with the same data logger used for data collection. Several filters, such as time-zero corrections, dewow, bandpass, gain, and background removal, are built into the software, which allows for the correction of basic interference such as background noise that occurs during GPR collection. These filters are applied to a batch of data for quick access to post-processed B-scans and C-scans. This first visualization helped in data quality assessment as well as in the identification of areas of interest.

Secondly, for advanced data visualization and further analysis, we used GPR-Slice v7.0. Apart from the same 3D imaging approach described in the previous paragraph, elevation values from the GNSS data were used for topographic corrections and the tilt of the antenna.

### 3. Conclusions

The combination of innovative hardware and software allowed the team to use GPR to locate the buried graves quickly, accurately map the position on the ground surface, and proceed to the excavation and successful recovery of the materials representing human remains.

GPR can be a superb tool to detect buried objects, but clear communication of the results and the possible limitations of the data collection are key to the use of the technology in real-world forensic applications. The forensic experts need to be able to clearly understand the GPR interpretations to give the geophysical information context during the search for clandestine graves.

The training exercise began with several limitations. Having actual human remains instead of plastic ones would have produced different GPR signal responses. Subsidence in the ground and the change in the soil conductivity due to decomposition would have improved GPR interpretation. The conditions were far from ideal for the GPR survey, as the ground contained a substantial amount of construction debris that produces significant reflections in the GPR data and can obscure anomalies created by the burial. Despite these challenges, the GPR results were able to narrow down the possible burial areas for testing by the training teams to eight locations and indicate the actual sites of the simulated clandestine burials.

Augmented reality (AR) was an impressively effective tool for crime scene investigations. The incorporation of AR was key in communicating the specific placement of the GPR anomalies indicating the location of the possible burial locations. This type of accessible software that can be used in the field is useful in planning successful excavations based on GPR results.

The conducted GPR research and the subsequent archaeological verification of its results, as part of a multi-disciplinary training exercise, indicate the enormous potential of this method to be used in Poland by forensic specialists and the Institute of National Remembrance. GPR was applied in in a strongly anthropogenically modified area, and the results were communicated by the GPR expert with teams consisting of law enforcement specialists, forensic specialists, and archaeologists, leading to a successful exercise in collaborative geophysical and forensic methodology.

**Author Contributions:** Conceptualization, T.B., F.C., J.F. and M.T.; methodology, F.C. and J.F.; software, F.C. and A.N.; writing—original draft preparation, T.B., F.C. and M.T.; writing—review and editing, T.B. and J.F.; visualization, F.C. and A.N.; supervision, T.B. and M.T.; project administration, T.B. and M.T.; funding acquisition, T.B. and M.T. All authors have read and agreed to the published version of the manuscript.

**Funding:** This research received no external funding.

**Institutional Review Board Statement:** Not applicable.

**Informed Consent Statement:** Not applicable.

**Data Availability Statement:** Not applicable.

**Conflicts of Interest:** The authors declare no conflict of interest.

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
