# Peer review of "A Controlled Experiment to Test the Efficacy of Ground-Penetrating Radar in the Search for Clandestine Burials in Poland"

_forensicsci, doi:10.3390/forensicsci2030043_

Round 1

Reviewer 1 Report

The paper presents results of GPR investigations for detection of clandestine burials. A significant amount of the work was conducted but the results presentation was done too superficially and I cannot evaluate its suitability for publication in Forensic Sciences without paper improvement. I would suggest to following:

1.1. The literature review can be extended and reflect the main publications in this area. For example, the paper  Berezowski et al., 2021 presents a good review. Detailed investigations of GPR applications for finding graves in a cemetery are published by Fernández-Álvarez et al.,2018 and Barone et al., 2016. The paper has to explain what are new results there?

 2.    The paper title is too wide. Term “GPR” can be included to the title.

 3.    The presented scans do not allow clear separation of the possible burial locations. I recommend to add description of scans (axes, color units) and to explain on what basis the decisions about possible burial locations were made?

 4.    Advanced data processing section is talking about “a combination of a novel web-based data analysis platform and traditional desktop software was used. First, GPR Insights allowed for quick 3D visualization from the field with the same data logger used for data collection. Several filters such as time-zero corrections, dewow, bandpass, gain, background removal, 2D migration and Hilbert transform together with slicing and gridding, using inverse distance interpolation algorithms, are built-in and applied in batch for quick access to post-processed B-Scans and C-Scans”.  This part could be a significant part of the paper. I recommend clear description of the each step with references and to show how the applied processing does improve the data visualization.

 5.    The paper mentioned about  Augmented Reality (AR) feature “as an impressive data visualization tool for crime scene investigations” but this feature and its application is not explained.

References

Barone, P.M., Swanger, K.J., Stanley-Price, N. and Thursfield, A., 2016. Finding graves in a cemetery: preliminary forensic GPR investigations in the Non-Catholic Cemetery in Rome (Italy). Measurement, 80, pp.53-57.

Berezowski, V., Mallett, X., Ellis, J. and Moffat, I., 2021. Using ground penetrating radar and resistivity methods to locate unmarked graves: a review. Remote Sensing, 13(15), p.2880

Fernández-Álvarez, J.P., Rubio-Melendi, D., Martínez-Velasco, A., Pringle, J.K. and Aguilera, H.D., 2016. Discovery of a mass grave from the Spanish Civil War using Ground Penetrating Radar and forensic archaeology. Forensic Science International, 267, pp.e10-e17.

Author Response

  1. The references have been extended
  2. The title has been changed
  3. The Figures were adjusted
  4. This part was rewritten
  5. An attempt was made to explain it better

Reviewer 2 Report

The article presents a technical report on practical activities in the use of geophysical methods in the search of clandestine grave, during the international workshop ‘Forensic Search and Recovery of Clandestine Graves’ held in Poland in September of 2021.

In general, the work presents a relevant contribution to the discussion of geophysical methods applied in forensic science. However, I believe that the work can be improved with the addition of a new section with a deeper discussion on the results, listing and discussing in greater detail the main difficulties faced during the study, and also putting some possible recommendations for a standardization of procedures in these studies.

In its current format it is an account of the activities carried out during the workshop, but with very limited critical discussion. I believe that this should be the biggest contribution in this type of work, and therefore my recommendation is for a revision of the work introducing these additional discussions in the text. The topic is very important and related to the special issue on "Forensic Geoscience and Death Investigations", so my final recommendation is for a review decision before possible publication.

Author Response

The discussion and conclusions have been deepened. Also the title and the introduction section have been changed and ajusted to the reviewers' comments. The use of English is better thanks to involvment of American archaeologist - Jamie Frattarelli.

Round 2

Reviewer 1 Report

I am satisfied by the paper improvement. 

Reviewer 2 Report

The authors have satisfactorily met my recommendations, and now the format and conclusions of the work are suitable for publication. My recommendation is for the acceptance of the work in its present form.